# Croissant: A Metadata Format for ML-Ready Datasets

**Mubashara Akhtar[1*], Omar Benjelloun[2*], Costanza Conforti[2*], Luca Foschini[3*],**
**Pieter Gijsbers[4], Joan Giner-Miguelez[5,22*], Sujata Goswami[6], Nitisha Jain[1*],**
**Michalis Karamousadakis[7], Satyapriya Krishna[8], Michael Kuchnik[9*], Sylvain Lesage[10*],**
**Quentin Lhoest[10*], Pierre Marcenac[2*], Manil Maskey[11], Peter Mattson[2], Luis Oala[12*],**
**Hamidah Oderinwale[13], Pierre Ruyssen[2*], Tim Santos[14], Rajat Shinde[15*], Elena Simperl[1,16*],**
**Arjun Suresh[17], Goeffry Thomas[2,18*], Slava Tykhonov[19*], Joaquin Vanschoren[4*],**
**Susheel Varma[3], Jos van der Velde[4*], Steffen Vogler[20], Carole-Jean Wu[9], Luyao Zhang[21]**
Authors in alphabetical order

[*]Core contributors [1]King's College London, [2]Google, [3]Sage Bionetworks, [4]Eindhoven University of Technology, [5]Universitat Oberta de Catalunya, [6]Oak Ridge National Laboratory, [7]Plaixus Ltd, [8]Harvard University, [9]Meta, [10]Hugging Face, [11]NASA, [12]Dotphoton, [13]McGill University, [14]Graphcore, [15]NASA IMPACT & UAH, [16]Open Data Institute, [17]GATE Overflow, India, [18]Kaggle, [19]DANS-KNAW, [20]Bayer, [21]Duke Kunshan University, [22]Barcelona Supercomputing Center (BSC)

## Abstract

Data is a critical resource for machine learning (ML), yet working with data remains a key friction point. This paper introduces Croissant, a metadata format for datasets that creates a shared representation across ML tools, frameworks, and platforms. Croissant makes datasets more discoverable, portable, and interoperable, thereby addressing significant challenges in ML data management. Croissant is already supported by several popular dataset repositories, spanning hundreds of thousands of datasets, enabling easy loading into the most commonly-used ML frameworks, regardless of where the data is stored. Our initial evaluation by human raters shows that Croissant metadata is readable, understandable, complete, yet concise.

## 1 Introduction

Recent machine learning (ML) advances highlight the critical role of data management in achieving technological breakthroughs. Yet, working with data remains time-consuming and painful due to a wide variety of data formats, the lack of interoperability between tools, and the difficulty of discovering and combining datasets [1, 2]. Data's prominent role in ML also leads to questions about its responsible use for training and evaluating ML models in areas such as licensing, privacy, or fairness, among others [3]. New approaches are needed to make datasets easier to work with, while also addressing concerns around their responsible use.

This paper[1] presents *Croissant*, a metadata format designed to improve ML datasets' discoverability, portability, reproducibility, and interoperability. Croissant makes datasets "ML-ready" by recording ML-specific metadata that enables them to be loaded directly into ML frameworks and tools (see Figure 2 for sample code). Croissant describes datasets' attributes, the resources they contain, and their structure and semantics. This uniform description streamlines their usage and sharing within the ML community and between ML platforms and tools while fostering responsible ML practices. Figure 1 gives an overview of the Croissant lifecycle and ecosystem.

Croissant can describe most types of data commonly used in ML workflows, such as images, text, audio, or tabular. While datasets come in a variety of data formats and layouts, Croissant exposes a unified "view" over these resources. It lets users add semantic descriptions and ML-specific information. The Croissant vocabulary [5] does not require changing the underlying data representation, and can thus be easily added to existing datasets, and adopted by dataset repositories.

---

[1]A shorter preliminary introduction to Croissant was presented at the DEEM 2024 workshop [4].

To assess Croissant's usability, we conducted a preliminary usability evaluation on metadata creation for language, vision, audio, and multi-modal datasets. Several practitioners annotated ten widely used ML datasets. We analyzed the consistency of their responses and collected their feedback on Croissant.

The remainder of the paper is structured as follows: in Section 2 we contextualize related work. In Section 3 we describe the Croissant format, its integrations, and the tools that support it. Section 4 comprises the user study and discusses its results and limitations.

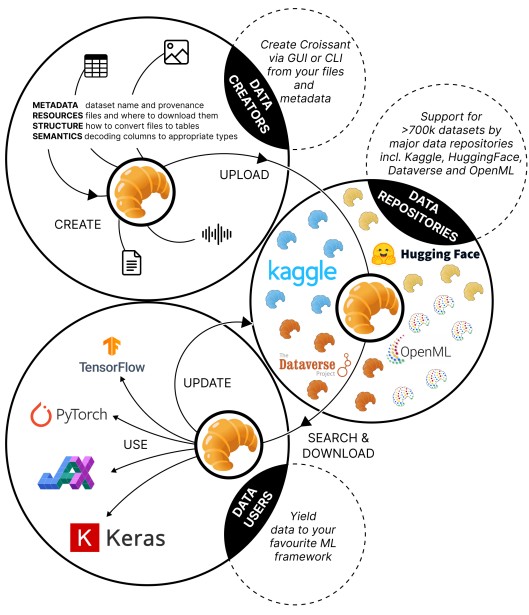

Figure 1: The Croissant lifecycle and ecosystem.

## 2 Related Work

While there have been many prior efforts in standardizing dataset metadata, they typically lack ML-specific support, do not work with existing ML tools, or lag behind the demands of dynamically evolving requirements, such as responsible ML. We outline the state of the field below.

**Vocabularies for Dataset Documentation.** Dataset documentation is indispensable for effective data management and serves as a foundational element for training and evaluating ML models [6]. Metadata descriptions of datasets enhance their discoverability, interoperability, and usability, which is critical for advancing research and data-driven applications. Ontologies and vocabularies are semantic web tools used to standardize dataset documentation. While vocabularies comprise sets of terms and their meanings to describe data consistently, ontologies provide a structured framework to define and relate these concepts within a domain. Ontologies and vocabularies are evaluated for their coverage (i.e., do they represent all relevant concepts), accuracy (correctness of definitions and relationships), consistency (no logical contradictions), and usability (ease of use and integration). This is done through methods like competency questions, expert validation, and use-case testing [7].

**Standards for Catalogs and Metadata.** With the increase of data availability online, various efforts have focused on making data both discoverable and user-friendly by supplementing datasets with comprehensive metadata. This metadata may include details about the dataset, such as authorship, format, and intended use, all structured consistently to support automated processing and retrieval. Key efforts towards documentation have led to the creation of standards like the Data Catalog Vocabulary (DCAT) [8] and the `Dataset` vocabulary in `schema.org` [9]. DCAT facilitates interoperability among web-based data catalogs, enabling users to aggregate, classify, and filter datasets efficiently. `Schema.org` [10] acts as a de facto standard for metadata, helping search engines discover and index published web content, including datasets, thus enhancing dataset accessibility and understandability. This versatility allows `schema.org` to describe a wide array of content types effectively. Other frameworks, such as Data Packages [11] and CSV on the Web [12] support methods for describing and exchanging tabular data. The Global Alliance for Genomics and Health's Data Use Ontology (DUO) [13] refines data usage terms with optional modifiers, improving clarity in genomic data sharing agreements. Efforts towards integration of FAIR principles (Findability, Accessibility, Interoperability, and Reusability) [14] in metadata vocabularies are also noteworthy. Despite their utility for specific domains and formats, these standards do not entirely meet the specialized needs of data management within the ML domain. In this context, the compliance of ML-ready datasets with the FAIR principles is a primary need for improving discoverability, portability and reproducibility in the ML ecosystem. The adoption of standard metadata description practices across the broader community further enhances the interoperability of ML datasets from diverse domains.

```
1  # 1. Point to a local or remote Croissant JSON file
2  import mlcroissant as mlc
3  url = "https://huggingface.co/api/datasets/fashion_mnist/croissant"
4  # 2. Inspect metadata
5  print(mlc.Dataset(url).metadata.to_json())
6  # 3. Use Croissant dataset in your ML workload
7  import tensorflow_datasets as tfds
8  builder = tfds.core.dataset_builders.CroissantBuilder(
9      jsonld=url, file_format="array_record")
10 builder.download_and_prepare()
11 # 4. Split for training/testing
12 train, test = builder.as_data_source(
13     split=["default[:80%]", "default[80%:]"])
```

Figure 2: Users can easily inspect datasets (e.g., Fashion MNIST [24]) and use them in data loaders with Croissant. See Supplementary material or visit https://github.com/mlcommons/croissant for more examples.

**Operationalizing Responsible ML through Data Work.** Data-centric ML [2, 15] is increasingly seen as critical to the development of trustworthy ML systems, including aspects such as fairness, accountability, transparency, data privacy and governance, safety, and robustness [16]. Seminal works, such as Datasheets for Datasets [6] and Data Statements [17], have emphasized the importance of dataset documentation to assess and increase the trustworthiness of ML systems. Several related documentation efforts such as Data Cards [18] and Data Nutrition Labels [19] have been inspired them. ML data repositories, such as Kaggle [20], OpenML [21] and Hugging Face [22], have initiated their own metadata documentation efforts. Hugging Face, for example, provides Dataset Cards [23] that include summaries, fields, splits, potential social impacts, and biases inherent in the datasets.

These approaches typically rely on data documentation written in natural language, without a standard machine-readable representation, which makes data documentation challenging for machines to read and process. Croissant fills this gap by providing a standardized framework for data documentation that ensures semantic consistency and machine readability, thereby facilitating seamless integration with existing tools and frameworks used by the ML community.

## 3  The Croissant Format

The Croissant format is a community-driven metadata vocabulary for describing datasets that builds on Schema.org [10]. Croissant is divided into four layers: *(i)* The *Dataset Metadata Layer*, containing relevant information such as name, description, and version. *(ii)* The *Resource Layer* describes the source data used in the dataset. *(iii)* The *Structure Layer*, describing and organizing the structure of the resources. *(iv)* The *Semantic Layer*, which provides ML-specific data interpretation and semantics. A more detailed description of the Croissant format can be found in the official specification [5]. Documentation and code is available online[2].

In the remainder of this section, we illustrate each layer with examples from popular ML datasets. Afterwards, we briefly describe the Croissant Responsible AI extension, and then provide an overview of ML frameworks, tools, and repositories that currently support Croissant.

### 3.1  The Dataset Metadata Layer

Croissant dataset descriptions, illustrated in Figure 3, are based on schema.org/Dataset, a widely adopted vocabulary for datasets on the Web [9], hence ensuring interoperability with existing standards and tools. Croissant specifies constraints on which schema.org properties are required, recommended and optional, and adds additional properties, e.g., to represent snapshots, live datasets, and citation information.

---

[2]https://docs.mlcommons.org/croissant/

```
1  {
2    "@type": "sc:Dataset",
3    "name": "PASS",
4    "dct:conformsTo":
        "http://mlcommons.org/croissant/1.0",
5    "description":
6    . "PASS is a large-scale image
       dataset...",
7    "citeAs": "@Article{asano21pass, ...",
8    "license": "cc-by-4.0",
9    "url":
        "https://www.robots.ox.ac.uk/.../pass/"
10
11   "distribution": [
12   {
13     "@id": "metadata",
14     "@type": "cr:FileObject",
15     "contentUrl":
         "https://zenodo.org/661...",
16     "sha256": "0b033707ea49365a5ffdd1461...",
17     "encodingFormat": "text/csv"
18   },
19   {
20     "@id": "pass0",
21     "@type": "cr:FileObject",
22     "contentUrl":
         "https://zenodo.org/661...",
23     "sha256": "0be3a104d6257d83296460b...",
24     "encodingFormat": "application/x-tar"
25   },
26   {
27     "@id": "image-files",
28     "@type": "cr:FileSet",
29     "containedIn": { "@id":"pass0" }
30     "includes": "*.jpg",
31     "encodingFormat": "image/jpeg"
32   }],
33  }
34
35
36
37
38
39
40
41
```

Figure 3: Dataset metadata and resources for the PASS dataset.

```
1  { "@id": "images",
2    "@type": "cr:RecordSet",
3    "key": "images/hash",
4    "field": [
5    { "@id": "images/image_content",
6      "@type": "cr:Field",
7      "dataType": "sc:ImageObject",
8      "source": {
9        "fileSet":{"@id": "image-files"},
10       "extract":{"fileProperty":"content"}
11     }
12   },
13   {
14     "@id": "images/hash",
15     "@type": "cr:Field",
16     "dataType": "sc:Text",
17     "source": {
18       "fileSet": {"@id": "image-files"},
19       "extract": {"fileProperty":
         "filename"},
20       "transform": {"regex":
         "([^\\/]*)\\.jpg"}
21     },
22     "references": {
23       "fileObject": {"@id": "metadata"},
24       "column": "hash"
25     }
26   },
27   { "@id": "images/coordinates",
28     "@type": "cr:Field",
29     "dataType": "sc:GeoCoordinates",
30     "subField": [
31     { "@id": "images/coordinates/latitude",
32       "@type": "cr:Field",
33       "source": {
34         "fileObject": {"@id": "metadata"},
35         "column": "latitude"}
36     },
37     { "@id": "images/coordinates/longitude",
38       "@type": "cr:Field",
39       "source": {
40         "fileObject": {"@id": "metadata"},
41         "column": "longitude"}
42     }]
43   }]
44  }
```

Figure 4: A `RecordSet` that joins images and structured metadata from the PASS dataset.

## 3.2 The Resources Layer

This layer represents the data resources (e.g., files) of the dataset. `Schema.org` properties are insufficient to adequately describe dataset contents with complex layouts, which are common for ML datasets. This layer provides two primitive classes to address this limitation and describe dataset resources: `FileObject` to describe individual files and `FileSet` to describe sets of files.

Figure 3 shows an excerpt of the Croissant definition of the PASS dataset [25], where declarations of object names are highlighted in yellow, with references in orange. This distribution includes two `FileObjects`: a CSV file containing metadata about the dataset (line 13) and an archive file containing images (line 20). Moreover, `FileSet` (in line 27) is used to refer to a collection of images, videos, or text files that contain the (unlabeled) data used for training and inference. Since there can be numerous files, `FileSets` are specified with inclusion/exclusion filters (e.g., a pattern matching all files that should be included) as shown on line 30.

## 3.3 The Structure Layer

While `FileObject` and `FileSet` describe a dataset's resources, they lack information on how the content of the resources is organized. This is addressed with `RecordSet`, which allows loading data of various formats into a standard representation, including structured (CSV and JSON) and

unstructured (text, audio, and video) data. Handling all data formatting information in one layer abstracts away format heterogeneity, addressing a key challenge in processing and loading ML data.

`RecordSet` provides a common structure description for records that may contain multiple fields, which can be used across different modalities. As an example, Figure 4 shows a `RecordSet` combining images from PASS with additional features from a metadata CSV file. Each `Field` in the `RecordSet` defines the source of its data, which may refer to the contents of elements in a `FileSet`. For instance, the `Field images/image_content` in line 9 refers to the `image-files FileSet` and also points to the specific property to extract in line 10.

`Fields` can be nested, as we can see in the `images/coordinates` field, which contains two sub-fields: `images/coordinates/latitude` and `images/coordinates/longitude`. Croissant supports nesting entire `RecordSets`, e.g., to add annotations (e.g. object bounding boxes) to images, where each image may correspond to multiple structured annotations. See Croissant's COCO [26] definition[3] for a representative example. `RecordSet` also supports joining heterogeneous data and data manipulation methods, like JSON Path and regular expressions, for flexible data extraction and transformation.

### 3.4 The Semantic Layer

The semantic layer introduces a number of useful features in the context of ML data. These are implemented using the primitives defined in the previous sections, generally as new classes or properties defined in the Croissant namespace. Semantic typing is used to describe important aspects of ML practice, such as the dataset splits (train, test, validation) as well as dataset labels. Additionally, semantic typing is used to describe commonly used data types, such as bounding boxes, categorical data, or segmentation masks. As an example, in Figure 4, the structured `Field images/coordinates` has the dataType `GeoCoordinates`[4] from schema.org. The subFields `images/coordinates/latitude` and `images/coordinates/longitude` are implicitly mapped to the latitude and longitude properties associated with that class, because their names match by suffix.

### 3.5 The Croissant-RAI Extension

Croissant-RAI [27] is an extension of the Croissant format that builds on existing responsible AI (RAI) dataset documentation approaches, such as Data Cards [18] and Datasheets for Datasets [6], making it easier to publish, discover, and reuse RAI metadata. The extension was developed around RAI use cases such as documenting the data life cycle, data labeling and participatory processes, information for AI safety, fairness assessments, and regulatory compliance. It was developed through a multi-step, iterative vocabulary engineering process. Based on the target use cases, a list of properties was defined through evaluation of related dataset documentation vocabularies and the Croissant vocabulary with an aim to detect overlaps and gaps. The resulting properties were evaluated by annotating example datasets to verify their usability and usefulness. For more details, see [28].

### 3.6 Croissant Tools and Integrations

In parallel with the definition of the Croissant format, we have pursued a number of integrations, with the goals of 1) making Croissant immediately useful to users, and 2) grounding Croissant in the requirements of real-world datasets and tools. Figure 1 gives an overview of the Croissant ecosystem.

**Data Repositories.** Croissant has been integrated into three major dataset repositories: Hugging Face Datasets, Kaggle Datasets, and OpenML, which together describe over 400,000 datasets in the Croissant format. This integration has succeeded with minimal effort because Croissant is an extension of the widely adopted `Schema.org/Dataset` vocabulary and does not require changing the existing data layout. Supporting Croissant involved adding additional fields to existing metadata. Furthermore, most repositories offer normalized data representations (Hugging Face and OpenML convert most datasets to Parquet) and their own data types (such as relational schemas for tabular data). Consequently, the conversion to Croissant primarily focuses on managing these data formats and specifying associated data types as `RecordSet` definitions.

---

[3] https://github.com/mlcommons/croissant/blob/main/datasets/1.0/coco2014/metadata.json
[4] http://schema.org/GeoCoordinates

In addition to the support from individual data repositories, Croissant is also supported by Google Dataset Search [29]. When a user searches for a query that returns Croissant datasets, a special filter allows them to restrict the results to only Croissant datasets. This functionality allows users to effectively search for Croissant datasets across data repositories and the entire web.

**ML Frameworks.** Croissant's reference implementation is a standalone Python librarythat supports the validation of Croissant dataset descriptions, their programmatic creation and manipulation, and serialization into JSON-LD. To consume data, the library provides an iterator abstraction that interoperates with existing data loaders. The TensorFlow Datasets [30] library provides a dataset builder[5] that prepares the dataset on disk in a format compatible with JAX, TensorFlow and PyTorch loaders. Alternatively, frameworks such as PyTorch DataPipes [31] interface with the Croissant library by wrapping the iterator directly. We anticipate that additional optimization opportunities will arise with more varied and larger datasets, perhaps requiring distributed execution as well as more advanced operator scheduling.

**Croissant Editor.** Croissant is primarily a machine readable format (in JSON-LD), so users may find it hard to create dataset descriptions by hand. We developed the Croissant Editor[6], (also on GitHub[7]), a tool that lets users visually create and modify Croissant datasets. The Croissant Editor provides form-based editing and validation of Croissant metadata, and bootstraps the definition of resources and `RecordSets` by inferring them from the data uploaded by the user. The editor integrates the Croissant Responsible AI extension, and guides users in describing RAI aspects of their datasets.

### 3.7 The Croissant Working Group

We designed the Croissant format in an open and participatory way. The MLCommons Croissant Working Group (WG)[8] consists of diverse stakeholders and domain experts from academia, industry, research organizations, and collaborative networks such as the AI for Public Good network. Use cases were discussed and presented to WG members (including domain experts) as they were developed, ensuring that diverse views and priorities were covered. The schema is designed to be modular and extensible, allowing for domain-specific attributes and concerns to be integrated into the Core Croissant format. We continuously collect feedback from working group members and users and are committed to incorporating this feedback in future versions of Croissant. Additionally, Croissant is based on `schema.org`, a well-established vocabulary.

## 4 Croissant Evaluation: A User Study with ML Practitioners

This section describes the user study we conducted to evaluate the Croissant metadata format. We asked machine learning practitioners to annotate a variety of datasets commonly used in the ML community. Human annotators authored a subset of the Croissant and Croissant-RAI attributes and assessed them based on criteria commonly used in vocabulary evaluation [32].

### 4.1 The User Study Process

**Recruitment of Annotators and Annotation Process.** We recruited nine volunteers from the Croissant development community who were all proficient in English with backgrounds in vocabulary and ontology engineering, dataset documentation, ML benchmarking, and responsible AI. We collected demographic information from all annotators, which we published in the user study report [33]. For each one of the ten datasets, we collected metadata definitions from three annotators, resulting in thirty annotations. Each human annotator assessed approximately three datasets on average, with three annotating one dataset and one person annotating six datasets.[9]

The instructions for the annotators were comprised of: $(i)$ a short introduction to the Croissant metadata format; $(ii)$ the purpose of the user study; $(iii)$ the definitions of the requested Croissant and Croissant-RAI attributes; $(iv)$ links to the format specifications, and $(v)$ a link to each dataset

---

[5]https://www.tensorflow.org/datasets/format_specific_dataset_builders#croissantbuilder
[6]https://huggingface.co/spaces/MLCommons/croissant-editor
[7]https://github.com/mlcommons/croissant/tree/main/editor
[8]See the MLCommons website for further details: https://mlcommons.org/working-groups/data/croissant/
[9]We publish the user study specifications and collected data [33].

| Criteria | Question | Answer Options |
|---|---|---|
| Answer Confidence | How confident are you that your provided annotations are correct? | **1** (no confidence) **5** (very confident that annotations are correct) |
| Dataset Understanding | How well did you understand the dataset (e.g. the task, domain, modality, etc.)? | **1** (I don't understand the dataset at all) - **5** (the dataset incl. its purpose, creation, etc. is very clear and understandable for me) |
| Completeness | Is there any (in your opinion important) information about the dataset which you can't define using Croissant? | **1** (yes, there is lots of critical information about the dataset that Croissant does not capture) - **5** (no, every important information about this dataset, which might be useful for ML users, is capture in Croissant attributes) |
| Conciseness | Did you find any attributes redundant and not definable for this dataset? | **1** (yes, there are lots of redundant attributes) - **5** (no, none of the attributes is redundant) |
| Readability | How intuitive are the attributes names for you? A name is not intuitive if you need to check the specification to understand the attribute's name? | **1** (not intuitive at all, for each single attribute I checked the specification to understand it) - **5** (very intuitive, based on the name I could understand the attribute very well) |
| Understandability | Rate the ease of understanding the Croissant specification. | **1** (Understanding the spec. was very hard) - **5** (the spec. is very easy to understand) |

Table 1: Post-annotation assessment: Criteria, corresponding questions, and answer scales.

in the Hugging Face repository. Prior to starting the user study, we obtained ethical clearance and informed annotators about the data being collected and its purpose. For each dataset, annotators filled out a provided JSON template with the sixteen attributes to complete. Afterwards, annotators answered questions about their level of understanding of the datasets (see Table 1), and indicated their confidence in the annotations they provided on a Likert scale [34] between 1 and 5. We followed previous research [35] suggesting that confidence ratings can serve as a tool to understand potential annotation inconsistencies. The user study began in April 2024 and lasted approximately five weeks.

**Selection of Croissant Attributes.** We selected ten attributes from Croissant's Dataset Layer (see Section 3.1) and six Croissant-RAI attributes (Section 3.5). We selected attributes that $(i)$ require manual specification, $(ii)$ can be defined by dataset users using the following resources: the dataset itself, a publication describing the dataset, and the Hugging Face dataset card if available, and $(iii)$ support the discoverability and reproducibility of datasets, along the lines of previous literature on improving dataset usability via documentation. For example, missing or limited descriptions of datasets reduce their discoverability and hinder practitioners from using the dataset as intended [36]. Moreover, lack of information on data reproducibility, e.g., about the data collection and curation process, also impacts the dataset's adoption in the ML community [36]. Table 2 and Table 3 list the attributes selected for this study.

**ML Datasets.** We selected commonly used ML datasets from the language, vision, and audio modalities, based on their popularity on the Hugging Face (HF) Datasets repository. We further filtered datasets to require $(a)$ a pre-existing Croissant description, $(b)$ a dataset card in HF Datasets, and $(c)$ a publication that describes the dataset creation process. Table 4 lists all datasets.

**Evaluation.** To evaluate the collected attribute annotations, we studied the provided answers and assessed the agreement among annotators. To measure agreement for textual attributes (e.g., `sc:description`), we calculated BLEU scores between attribute annotations, which were in textual form and did not allow for inter-annotator agreement scores commonly used for measuring agreement based on categorical data. The BLEU metrics [37] measure text similarity based on overlapping 4-grams in text pairs. The score can be between $[0, 1]$ with 1 indicating perfect match between both compared texts. Hence, a score closer to one indicates higher agreement among all three annotations available for the respective attribute and dataset.

### 4.2 Mapping Evaluation Criteria to Croissant

Previous literature proposes different criteria for vocabulary evaluation [32, 48]. Following prior work [48], we evaluate Croissant on the five criteria we outline below. We further discuss how the criteria translate to Croissant and specify questions to evaluate each criterion in the context of our user study.

| Property |
|---|
| sc:description |
| sc:license |
| sc:name |
| sc:url |
| sc:creator |
| sc:publisher |
| sc:datePublished |
| sc:inLanguage |
| cr:citeAs |
| cr:isLiveDataset |

| Property | RAI Use Case |
|---|---|
| rai:dataCollection | Data life cycle |
| rai:dataCollectionTimeframe | Data life cycle |
| rai:dataAnnotationPlatform | Data labelling |
| rai:annotatorDemographics | Data labelling |
| rai:dataUseCases | AI safety and fairness evaluation |
| rai:personalSensitiveInformation | Compliance |

| Dataset | Modality |
|---|---|
| MMLU [38] | Language |
| Dolly-15k [39] | Language |
| FLORES [40] | Language |
| CIFAR10 [41] | Vision |
| MSCOCO [42] | Vision |
| Visual Genome [43] | Vision |
| MMMU [44] | VL |
| MathVista [45] | VL |
| MLS_Eng [46] | Audio |
| librispeech_asr [47] | Audio |

Table 2: Annotated Croissant attributes.  Table 3: Annotated Croissant-RAI attributes.  Table 4: Annotated datasets.

(1) **Consistency.** The criterion evaluates if a vocabulary is consistent and free of contradictions in its attribute definitions [32]. To measure Croissant's consistency, we studied how well annotations by different annotators for the same attribute and dataset aligned, i.e. based on the agreement among annotators.

(2) **Completeness.** A vocabulary is complete if it covers the specified intent. While Croissant is an ongoing effort and not fully complete, we evaluated during the user study if Croissant currently misses any attributes necessary to capture important information about commonly used ML datasets. We asked annotators to flag any important information about the datasets they annotated that could not be defined using Croissant.

(3) **Conciseness.** The conciseness criterion assesses whether a vocabulary avoids useless definitions and is free of redundancies. We measured this by asking annotators if they found any Croissant attributes redundant or not definable for the studied ML datasets.

(4) **Readability.** The readability criteria assess how intuitive the attribute names are. After completing the annotations, we asked annotators to indicate on a Likert scale of 1 to 5 how intuitive they found Croissant attribute names to be.

(5) **Understandability.** The understandability criteria evaluates how easily user can understand Croissant attributes from the provided documentation. During our user study, we instructed annotators to use the Croissant specifications [5, 27] and prompted them afterwards with questions.

### 4.3 Results and Discussion

This section analyses data collected during the user study. First, we evaluate the answers to the questions listed in Table 1. Second, we study the annotation of Croissant and Croissant-RAI attributes.

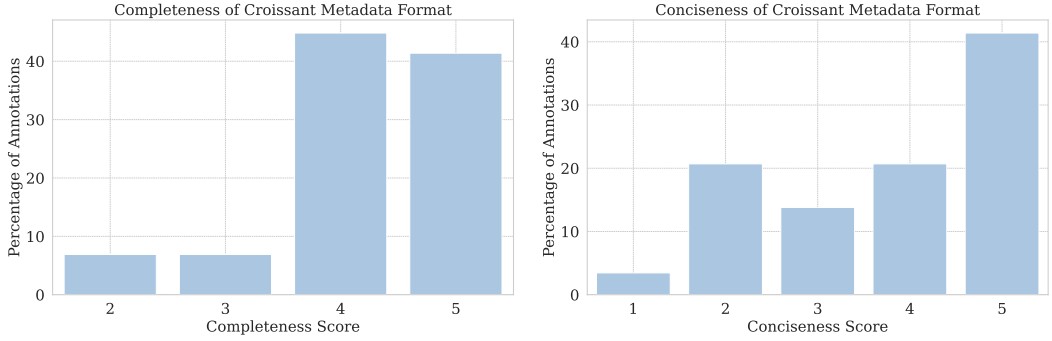

Figure 5: Answers to the *completeness* question.  Figure 6: Answers to the *conciseness* question.

**Criteria Evaluation.** Assessing annotators' ratings for the criteria in Table 1, we find that for over 80% of annotations (25 out of 30 annotations), Croissant attributes capture important information about the datasets (see Figure 5). For the conciseness criteria, we found a higher variance in ratings. While most annotations state that none or few requested attributes are redundant for the dataset (approx. 60%), seven annotations (around 23%) state that some attributes are redundant

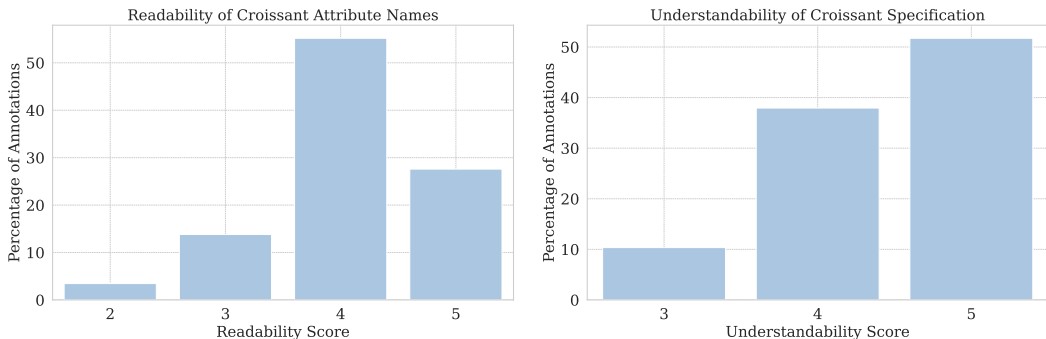

Figure 7: Answers to the *readability* question.    Figure 8: Answers to *understandability* question.

or not definable. This was due to Croissant-RAI attributes, e.g., `rai:annotatorDemographics` and `rai:personalSensitiveInformation`, which was either missing from the dataset's documentation or difficult to extract it. The majority of annotations (around $83\%$) found Croissant and Croissant-RAI attribute names intuitive, resulting in high *readability* scores (Figure 7). However, one attribute pair commonly confused annotators: `sc:creator` and `sc:publisher`. These attributes come from the schema.org vocabulary, which Croissant builds upon, and are already widely used to describe datasets on the Web. On the bright side, around $90\%$ of annotations stated that the specifications were understandable(Figure 8).

In addition to criteria-related questions, we asked annotators how confident they were regarding the correctness of their annotations and their understanding of the datasets. For the majority of annotations (more than $75\%$), their annotators selected that they were very confident. We found that the annotations with moderate confidence were indeed the ones with lower agreement on attribute values. Moreover, for the majority of annotations, the annotators selected that they had a clear understanding of the datasets, with around $80\%$ selecting four or five on the five point scale (see Table 1), which gave us strong confidence in the data collected through this study.

**Attributes Evaluation.**    Table 5 provides BLEU scores [37] as a measure of agreement for annotated text attributes. Overall, the average BLEU scores for Croissant attributes ($0.55$) is higher than for Croissant-RAI attributes ($0.41$). This difference can be attributed to several factors. First, multiple RAI attributes require a free-form text answer, which is more likely to differ across annotations than categorical or short-answer attributes such as `sc:name`, `sc:url`, or `sc:inLanguage`. Second, Croissant attributes are more easily extractable from the dataset's page on Hugging Face or from the introduction of the corresponding publication, while Croissant-RAI attributes often require detailed studying of the publication to find relevant RAI information, such as demographic information.

Attributes' average BLEU scores also diverges based on their expected values. Attributes with `Text` as the expected value have an average BLEU score of $0.4$ while `Date/Datetime` attributes have an average score of $0.47$.[10] Attributes with predefined values such as `Language` or `Url`, have an average score of $0.59$, indicating higher agreement. For example, comparing annotations across attributes, we observe the highest BLEU scores for `sc:license`. This is largely attributable to the fact that, while being free-form text, there is less variety in the attribute's annotations and therefore more matching 4-grams. The low BLEU score for the MathVista dataset is due to one annotator providing the text of the license instead of its name, as instructed in the specification [5].[11]

## 5    Limitations and Future Work

**Croissant Format.**    While the Croissant metadata format provides a shared representation across various ML tools, platforms, and frameworks, certain challenges remain that should be addressed in future work. First, its structure may pose difficulties for users unfamiliar with the format, potentially hindering broader adoption. In the future, we plan to extend Croissant tools (e.g., the Croissant editor) and provide comprehensive documentation, as these are the primary means of making Croissant

---

[10]See `https://schema.org/Text` and `https://schema.org/Date` for the exact definitions
[11]For future Croissant versions, we plan to formalize some free text attributes.

| Dataset | desc | lic | url | creator | publ | datePub | lang | citeAs | dataCol | time | plat | demogr | useCases | persInfo |
|---|---|---|---|---|---|---|---|---|---|---|---|---|---|---|
| flores | 0.03 | 0.6 | 0.45 | 0.88 | 0.54 | 0.12 | 0.84 | 0.31 | 0.4 | 0.08 | 0.34 | 0.0 | 0.42 | 0.0 |
| cifar-10 | 0.39 | 1.0 | 0.31 | 0.17 | 0.16 | 0.14 | 1.0 | 0.26 | 0.35 | 0.0 | 1.0 | 0.0 | 0.29 | 1.0 |
| dolly-15k | 0.56 | 1.0 | 1.0 | 0.82 | 0.5 | 0.28 | 0.34 | 0.75 | 0.57 | 0.0 | 1.0 | 0.0 | 0.39 | 0.01 |
| mscoco | 0.7 | 1.0 | 0.65 | 0.26 | 0.0 | 0.24 | 1.0 | 0.0 | 0.32 | 1.0 | 0.78 | 0.0 | 0.88 | 0.0 |
| visual gen | 0.41 | 1.0 | 0.18 | 0.49 | 0.0 | 0.51 | 1.0 | 1.0 | 0.29 | 0.19 | 0.0 | 0.84 | 0.27 | 1.0 |
| mmmu | 0.89 | 0.49 | 1.0 | 0.76 | 0.33 | 0.21 | 1.0 | 1.0 | 0.77 | 1.0 | 0.0 | 0.05 | 0.48 | 0.62 |
| mmlu | 0.13 | 0.0 | 0.56 | 0.97 | 0.37 | 0.32 | 1.0 | 0.79 | 0.6 | 1.0 | 0.07 | 0.65 | 0.45 | 0.0 |
| mathvista | 1.0 | 0.34 | 0.57 | 0.53 | 0.07 | 0.26 | 0.13 | 1.0 | 0.16 | 1.0 | 0.0 | 0.05 | 0.22 | 1.0 |
| mls_eng | 0.35 | 1.0 | 1.0 | 0.64 | 0.35 | 0.56 | 0.03 | 1.0 | 0.3 | 1.0 | 0.0 | 1.0 | 0.36 | 0.0 |
| librispeech | 0.73 | 1.0 | 0.17 | 0.82 | 0.33 | 0.44 | 1.0 | 0.04 | 0.34 | 1.0 | 0.0 | 0.29 | 0.25 | 0.21 |
| Average | 0.52 | 0.74 | 0.59 | 0.63 | 0.26 | 0.31 | 0.73 | 0.62 | 0.41 | 0.63 | 0.32 | 0.29 | 0.4 | 0.38 |
| Median | 0.52 | 1.0 | 0.57 | 0.64 | 0.33 | 0.28 | 1.0 | 0.75 | 0.35 | 1.0 | 0.07 | 0.05 | 0.39 | 0.21 |

Table 5: BLEU scores for annotated datasets and attributes (i.e. **desc**ription, **lic**ense, url, creator, **publ**isher, **datePub**lished, in**Lang**uage, citeAs, **dataColl**ection, dataCollection**Time**frame, dataAnnotation**Plat**form, annotator**Demogr**aphics, data**UseCases**, **pers**onalSensitive**Info**rmation)

datasets easier for users to utilize. This includes adding additional annotated dataset examples in the Croissant repository and developing community guidelines that account for domain-specific needs. Second, the Croissant editor, as an interface for users to create Croissant metadata, is still in its early stages and will be further developed and enhanced in future work. For example, it will support additional functionalities such as archiving files, nested fields, and more. Finally, as part of an ongoing effort to enhance and demonstrate Croissant's handling of complex data structures, a GeoSpatial extension for Croissant (named Geo-Croissant) will be released in the near future, accompanied by examples that demonstrate the handling of file formats such as HDF5 and Zarr.

**User Study.** While the user study presented here provides some initial insights on the usability of the Croissant vocabulary, it has a number of limitations that warrant follow-on work: ($i$) Increase the number of participants and annotated datasets, either by recruiting participants with the right combination of skills in dataset documentation and machine learning, or by encouraging the authors of datasets to create annotations directly, as they are most knowledgeable. Moreover, as the annotators were drawn from the Croissant community, this may have introduced bias as they were are familiar with the Croissant framework to some extent. Hence, a required future direction is exploring Croissant usage by non-expert users. ($ii$) Extend the evaluation results, as participants only focused on a subset of the attributes in the Croissant and Croissant RAI vocabularies. ($iii$) Finally, BLEU scores are a noisy metric for the quality of attribute annotations. However, due to the nature of our collected data being textual rather than categorical, standard agreement metrics such as Fleiss' Kappa [49] were not applicable. A possible future direction is to compare annotations with golden data from existing Croissant descriptions for the datasets.

## 6   Conclusion

This paper introduces Croissant, a metadata format for ML datasets. Croissant improves the discoverability, portability, and interoperability of ML datasets across data repositories, ML tools, frameworks, and platforms. The Croissant format addresses key challenges in data management by providing a standardized data representation, making datasets more discoverable, portable, and interoperable.

Croissant has already seen rapid adoption by popular ML dataset repositories and frameworks, and is recommended as a data artifact in the NeurIPS Datasets and Benchmarks Track. Moreover, Croissant metadata is deemed readable, understandable, complete, and concise by human raters. Still, Croissant's success will ultimately depend on further adoption in ML research and industry, the widespread availability of Croissant datasets, and support from ML tools and frameworks, thus we warmly invite further ML platform and tool developers to join the Croissant community.

Finally, Croissant's extendable nature and the broad range of datasets it can represent enables other communities to extend Croissant for their specific needs, similar to the Croissant-RAI extension developed for Responsible AI, and streamline collaborations between ML and other fields.

## Acknowledgments

This work was partly funded by the HE project MuseIT, which has been co-founded by the European Union under the Grant Agreement No 101061441. MuseIT has supported the work of Nitisha Jain. Views and opinions expressed are, however, those of the authors and do not necessarily reflect those of the European Union or European Research Executive Agency. Joan Giner-Miguelez is supported by the AIDOaRt project, which is funded by the ECSEL Joint Undertaking (JU) under grant agreement No 101007350. The JU receives support from the European Union's Horizon 2020 research and innovation programme and Sweden, Austria, Czech Republic, Finland, France, Italy, and Spain. Pieter Gijsbers, Joaquin Vanschoren, and Jos van der Velde would like to acknowledge funding by EU's Horizon Europe research and innovation program under grant agreement No. 952215 (TAILOR) and No. 101070000 (AI4EUROPE).

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
