## 7 Appendix

### 7.1 Code Examples

In this section, we illustrate through examples how users can work with Croissant files in their ML workflows, using the `mlcroissant` and TFDS libraries.

#### 7.1.1 Loading a Dataset from a Croissant File

```python
import mlcroissant as mlc
ds = mlc.Dataset("https://raw.githubusercontent.com/mlcommons/croissant/
    main/datasets/1.0/gpt-3/metadata.json")
metadata = ds.metadata.to_json()
print(f"{metadata['name']}:{metadata['description']}")
for x in ds.records(record_set="default"):
    print(x)
```

#### 7.1.2 Loading data from a Croissant JSON-LD file in an ML workflow by using TFDS

```python
import tensorflow_datasets as tfds
builder = tfds.dataset_builders.CroissantBuilder(
    jsonld="https://raw.githubusercontent.com/mlcommons/croissant/main/
    datasets/0.8/huggingface-mnist/metadata.json",
    file_format="array_record",
)
builder.download_and_prepare()
ds = builder.as_data_source()
print(ds["default"][0])
```

#### 7.1.3 Using Croissant into ML-workflow by loading into TFDS Data loader for HF Datasets

```python
# 1. Point to a local or remote Croissant file
import mlcroissant as mlc
url = "https://huggingface.co/api/datasets/fashion_mnist/croissant"

# 2. Inspect metadata
print(mlc.Dataset(url).metadata.to_json())

# 3. Use Croissant dataset in your ML workload
import tensorflow_datasets as tfds
builder = tfds.core.dataset_builders.CroissantBuilder(
    jsonld=url,
    record_set_ids=["record_set_fashion_mnist"],
    file_format="array_record",
)
builder.download_and_prepare()

# 4. Split for training/testing
train, test = builder.as_data_source(
    split=["default[:80%]", "default[80%:]"]
)
```

#### 7.1.4 Visualizing Bounding Boxes in Croissant using the COCO 2014 dataset

```python
# 1. Importing mlcroissant Python package
import mlcroissant as mlc

```

```python
# 2. Create a subset of the COCO 2014 dataset which offers bounding box
    annotations
record_set = "images_with_bounding_box"

# We download resources from the validation split to download smaller
    files.
distribution = [
    mlc.FileObject(
        id="annotations_trainval2014.zip",
        name="annotations_trainval2014.zip",
        description="",
        content_url=(
        "http://images.cocodataset.org/annotations/
    annotations_trainval2014.zip",
    ),
    encoding_format="application/zip",
        sha256="031296
    bbc80c45a1d1f76bf9a90ead27e94e99ec629208449507a4917a3bf009",
    ),
    mlc.FileObject(
        id="annotations",
        name="annotations",
        description="",
        contained_in=["annotations_trainval2014.zip"],
        content_url="annotations/instances_val2014.json",
        encoding_format="application/json",
    ),
]

# The record set has the `image_id` and the `bbox` (short for bounding
    box).
record_sets = [
    mlc.RecordSet(
        id="images_with_bounding_box",
        name=record_set,
        fields=[
            mlc.Field(
                id="images_with_bounding_box/image_id",
                name="image_id",
                description="",
                data_types=mlc.DataType.INTEGER,
                source=mlc.Source(
                    file_object="annotations",
                    extract=mlc.Extract(
                        json_path="$.annotations[*].image_id"
                    ),
                ),
            ),
            mlc.Field(
                id="images_with_bounding_box/bbox",
                name="bbox",
                description="",
                data_types=mlc.DataType.BOUNDING_BOX,
                source=mlc.Source(
                    file_object="annotations",
                    extract=mlc.Extract(
                        json_path="$.annotations[*].bbox"
                    ),
```

```python
57                ),
58              ),
59          ],
60      ),
61  ]
62
63  metadata = mlc.Metadata(
64      name="COCO2014",
65      url="https://cocodataset.org",
66      distribution=distribution,
67      record_sets=record_sets,
68  )
69
70  # 3. Creating the Croissant JSON-LD file
71  jsonld = epath.Path("croissant.json")
72  with jsonld.open("w") as f:
73      f.write(json.dumps(metadata.to_json(), indent=2))
74
75  # 4. Getting the first record from the generated Croissant JSON-LD
76  dataset = mlc.Dataset(jsonld=jsonld)
77  records = dataset.records(record_set=record_set)
78  record = next(iter(records))
79  print("The first record:")
80  print(json.dumps(record, indent=2))
81
82  # 5. Visualizing the bounding box
83  image_id, bbox = record["images_with_bounding_box/image_id"], record["
        images_with_bounding_box/bbox"]
84  url = f"http://images.cocodataset.org/val2014/COCO_val2014_{image_id:012d
        }.jpg"
85
86  # Download the image
87  print(f"Downloading {url}...")
88  response = requests.get(url)
89  image = Image.open(io.BytesIO(response.content))
90  draw = ImageDraw.Draw(image)
91
92  # COCO uses the XYWH format. PIL uses the XYXY format.
93  x1, y1, w, h = bbox
94  draw.rectangle((x1, y1, x1 + w, y1 + h), outline=(0, 255, 0), width=2)
95  display(image)
```

## 7.2 Croissant Health Metrics

Croissant Health is a framework to automatically scrape and compute metrics about Croissant from online dataset repositories. It has been implemented so far for Hugging Face Datasets and OpenML, and can be easily extended to new repositories. The metrics are derived from the crawl responses for hosted datasets and the number of `FileObjects`, `FileSets`, `RecordSets`, and `Fields` they contain. More detailed statistics will be added in the future.

### 7.2.1 Croissant Statistics for Hugging Face Datasets

Figure 9 shows that the number of successfully downloaded Croissant datasets from Hugging Face is over 100k, and the rate of invalid Croissant files is 25%. These statistics are key to identify issues with Croissant generation and fix errors. Figure 10 gives an idea of the shape of these datasets: On average, datasets are small across all dimensions, with less than 10 resources, `RecordSets`, and `Fields`.

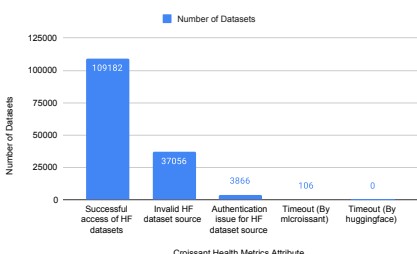

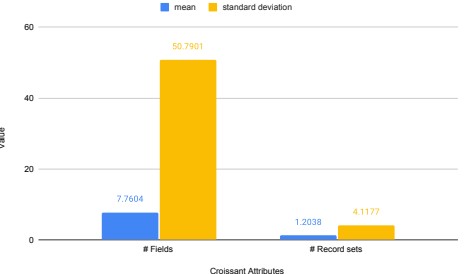

Figure 9: Scraping results for Croissant files of Hugging Face Datasets.

Figure 10: Illustration showing statistics for mean and standard deviation for the Croissant files hosted on Hugging Face datasets.

### 7.2.2 Croissant Stastistics for OpenML Datasets

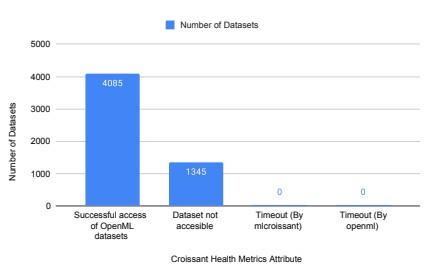

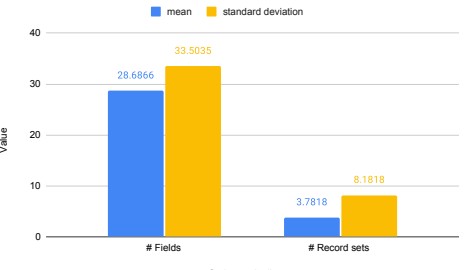

Figure 11: Scraping results for Croissant files hosted on OpenML.

Figure 12: Illustration showing statistics for mean and standard deviation for Croissant files hosted on OpenML datasets.

Figure 11 shows the Croissant adoption for the OpenML datasets and Figure 12 illustrates the statistics for OpenML datasets. The number of datasets is much smaller overall, at about 4k datasets. The rate of invalid Croissant files is around 25% due to authentication issues occurring while trying to access private datasets. We use Croissant Health[12] to monitor the health of the Croissant ecosystem by crawling online JSON-LD files shared across repositories. Currently, Croissant Health performs this check for Hugging Face and OpenML datasets only, but will be extended in future to further repositories.[13]

Figure 12 shows that these datasets are much more complex, with many datasets having a larger number or `FileObjects`, `RecordSets`, and `Fields` compared to the Hugging Face ones.

### 7.3 User Study

Figure 13 shows the participants' confidence in the annotations they provided, on a scale of 1 to 5. The majority of participants picked 4, which shows a high level of confidence in their ability to create Croissant metadata. It's interesting to contrast this number with the participants' level of understanding of datasets (Figure 14), which varies more broadly between 3 and 5.

Figure 15 gives an overview of how much time participants took for the user study. The majority of participants took 15-30 minutes to create the Croissant description of a dataset, which seems like a reasonable amount of time.

Finally, Figures 16, 17, and 18 show the instruction provided to participants for the user study.

---

[12]https://github.com/mlcommons/croissant/tree/main/health

[13]See https://github.com/mlcommons/croissant/blob/main/health/visualizer/report_openml.ipynb for further details.

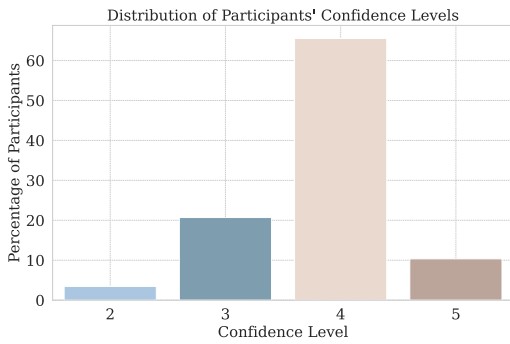

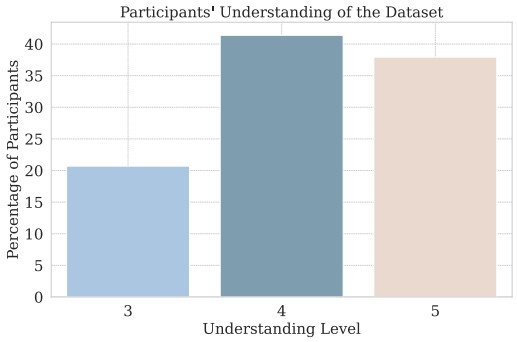

Figure 13: Annotators' confidence in provided annotations on a Likert scale from one to five. One indicates no confidence and five very high confidence in correct annotations.

Figure 14: Annotators' understanding of datasets on a Likert scale from one to five. One indicates that the annotator has no understanding of the dataset while five means that the annotator understands the dataset, including its purpose, creation, etc.

Time Taken to Annotate Datasets

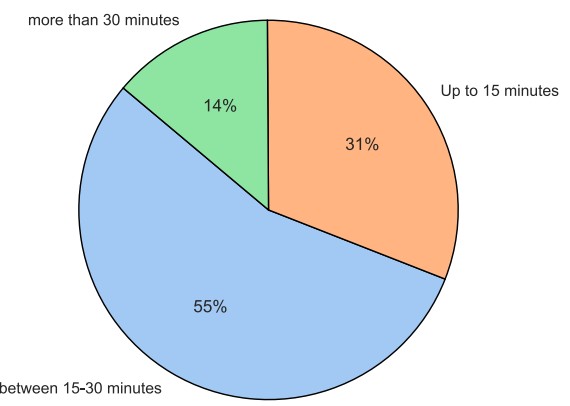

Figure 15: Time to create a Croissant description for a dataset.

## Croissant Manual Evaluation

**(Takes approx. 15 minutes to complete BUT as you start annotating, please note down the time as you will be asked afterwards how long it took to complete the annotations.)**

Croissant is a metadata format that simplifies the integration and use of datasets across diverse machine learning frameworks such as PyTorch, TensorFlow, and JAX. It provides a structured vocabulary for dataset attributes, facilitating the seamless exchange and utilization of datasets, which addresses challenges in discoverability, portability, reproducibility, and responsible AI (RAI).

Thank you for participating in this manual evaluation of the Croissant format. Your expertise and insights are crucial to enhancing its effectiveness and usability across various platforms. As part of this evaluation, you will be assigned one dataset from the HuggingFace datasets library. After accessing and reviewing the dataset, you will be asked to annotate it according to the 16 specified attributes listed in the instruction manual.

Your contributions are invaluable to us, and we appreciate your efforts in helping improve the Croissant metadata format for the wider machine learning community.

### INSTRUCTIONS ANNOTATION PROCESS

1. **First, look at the following attributes below to understand what information is requested from you about the dataset:**
   a. Croissant Core attributes to Evaluate:

   1. sc:description - Description of the dataset.
   2. sc:license - License details of the dataset; preferably a URL from a recognized source like SPDX.
   3. sc:name - The official name of the dataset.
   4. sc:url - URL where the dataset can be accessed.
   5. sc:creator - The creator(s) of the dataset, which can be an organization or person.
   6. sc:publisher - The publisher of the dataset, possibly distinct from the creator.
   7. sc:datePublished - The publication date of the dataset.
   8. sc:inLanguage - The language(s) in which the dataset content is available.
   9. cr:citeAs - Recommended citation for the dataset.
   10. cr:isLiveDataset - Indicator of whether the dataset is actively maintained and updated.

Figure 16: Instruction provided to user study participants for annotating ML datasets with selected Croissant/Croissant-RAI attributes (1/3).

11. rai:dataCollection - Description of the data collection process.
12. rai:dataCollectionTimeframe - Start and end date of the data collection process.
13. rai:dataAnnotationPlatform - Platform or tool used for data annotation.
14. rai:annotatorDemographics - Demographics of the annotators involved in the dataset labeling.
15. rai:dataUseCases - Potential uses of the dataset for AI safety and fairness evaluation.
16. rai:personalSensitiveInformation - Details about any personal sensitive information included in the dataset.

**2. Second, if you need further details to understand the attributes check out the resources below:**

- Croissant attributes documentation:
  https://mlcommons.github.io/croissant/docs/croissant-spec.html#dataset-level-information
- Croissant responsible AI (RAI) attributes documentation:
  https://mlcommons.github.io/croissant/docs/croissant-rai-spec.html#rai-property-information

**3. Third, checkout the following dataset:**

Dataset for Evaluation: https://huggingface.co/datasets/cais/mmlu

**4. Search if a publication or Huggingface dataset card is available describing the dataset. If so, take a look at the publication and/or dataset card, focusing on the dataset description and its creation process.**

Enter links to all external resources you used below:

**5. Complete the JSON template below, describing all sixteen attributes for the dataset you find under (3.)**

Please consider the following:
- To fill the attributes only use the dataset (see HuggingFace link above) and dataset documentations you find (e.g. a research paper or website describing the dataset)
- Do not use the Croissant metadata file of the dataset if one is available (e.g. on HuggingFace)
- If an attribute is not applicable to the dataset, enter "**NA**" as value.
- If an attribute is applicable but no information is provided in the dataset or paper, enter "**Unknown**".

Figure 17: Instruction provided to user study participants for annotating ML datasets with selected Croissant/Croissant-RAI attributes (2/3).

Fill all requested attributes in the JSON snippet below:

```json
{
// Croissant core attributes
    "sc:description": "",
    "sc:license": "",
    "sc:name": "",
    "sc:url": "",
    "sc:creator": "",
    "sc:publisher": "",
    "sc:datePublished": "",
    "sc:inLanguage": "",
    "cr:citeAs": "",
    "cr:isLiveDataset": "",
// RAI attributes
    "rai:dataCollection": "",
    "rai:dataCollectionTimeframe": "",
    "rai:dataAnnotationPlatform": "",
    "rai:annotatorDemographics": "",
    "rai:dataUseCases": "",
    "rai:personalSensitiveInformation": ""
}
```

**6. Enter (contact) information below and answer the questions:**

https://forms.gle/5ED5YmzXokSNCgTM9

Figure 18: Instruction provided to user study participants for annotating ML datasets with selected Croissant/Croissant-RAI attributes (3/3).

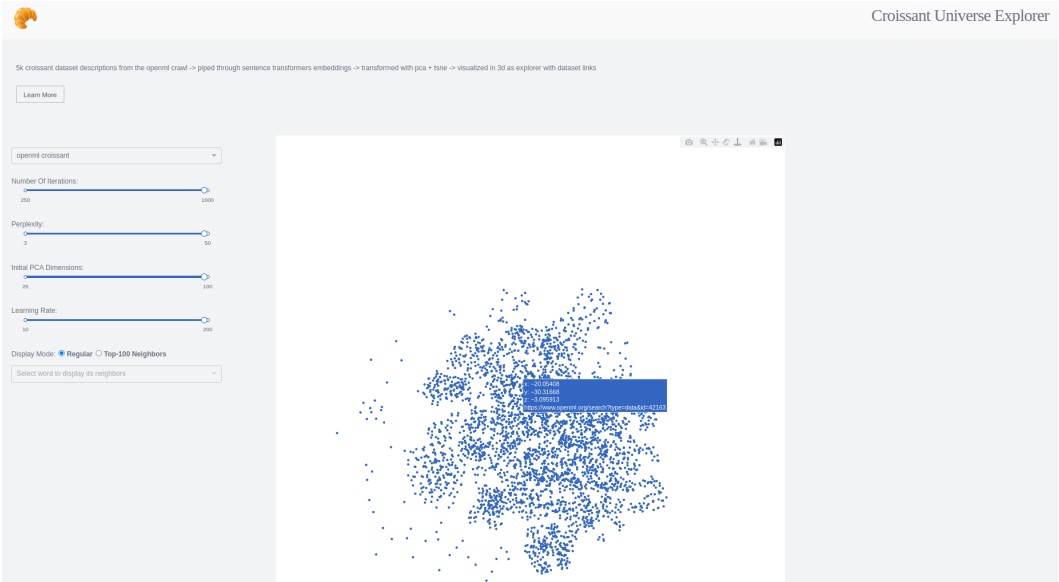

Figure 19: A visualizer example for exploring semantic similarity between datasets based on Croissant dataset Transformer and t-SNE embedding.

## 7.4 Semantic search with Croissant

The unified format of Croissant data makes it possible to scrape them from across the web and then conveniently embed and project them through a pipeline of your choice for semantic search among datasets. We provide a starter kit with an example of OpenML data at this address `https://github.com/mlcommons/croissant/tree/main/health/visualizer/explorer` where we

1. Scrape Croissant files from the OpenML API following the steps under `https://github.com/mlcommons/croissant/tree/main/health`
2. Read all Croissant dataset descriptions from the OpenML crawl (>5k)
3. Extract dataset descriptions and urls from the Croissant files
4. Project dataset descriptions onto an embedding space with a sentence transformer encoder
5. Project embeddings to a three-dimensional space with PCA and t-SNE
6. Explore semantic proximity of datasets in t-SNE embedding space

An example visualizer can be found on https://docs.mlcommons.org/croissant/.

## 7.5 Croissant Editor for Dataset Authors

The Croissant open-source editor (Figure 20) is a tool for generating Croissant metadata for dataset publishers. The editor abstracts away the details of the Croissant syntax via a familiar user interface. Users can drag-and-drop files to start creating a Croissant dataset.

The editor infers the resources and structure definitions from the data, and guides them in filling out required and optional fields (Figure 21). The editor can be run locally as well as on the Hugging Face interface and incorporates Croissant Core and Croissant RAI attributes (Figure 21) for generating Croissant file while hosting a dataset [14].

---

[14]https://HuggingFace.co/spaces/MLCommons/croissant-editor

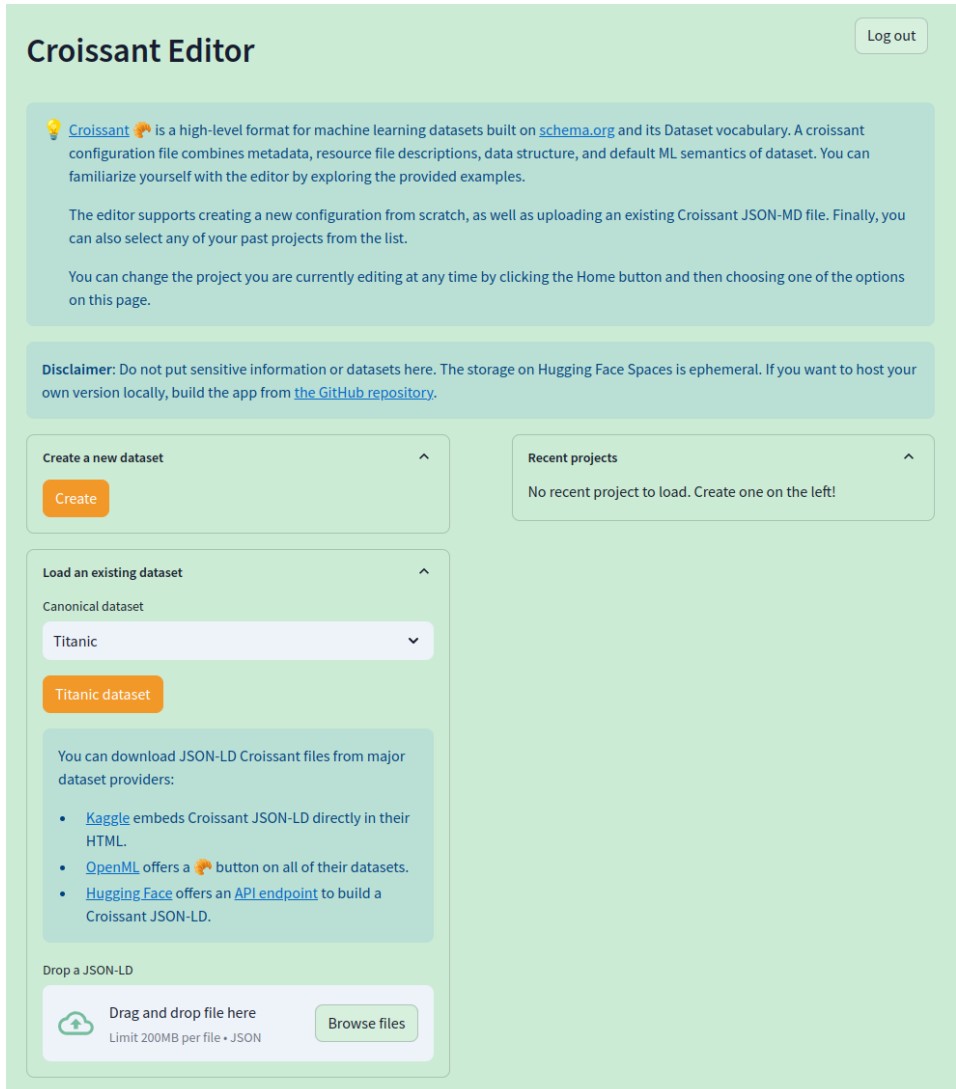

Figure 20: The Croissant editor Graphical User Interface (GUI).

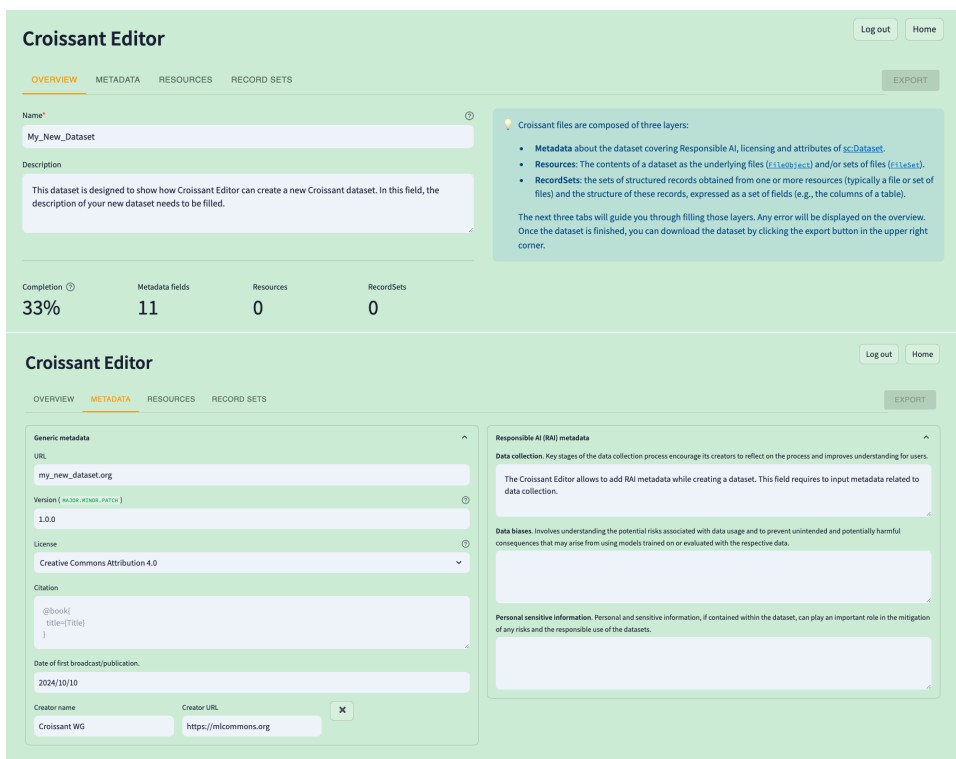

Figure 21: Illustration of the Croissant editor GUI for filling dataset and Responsible AI (RAI) attributes.