# OpenReview forum: "Croissant: A Metadata Format for ML-Ready Datasets"
_NeurIPS.cc/2024/Datasets_and_Benchmarks_Track — NeurIPS 2024 Track Datasets and Benchmarks Spotlight_

### Official Review · Reviewer_fBtg · 2024-07-24
**A valuable contribution to ML-dataset documentation infrastructure with a flawed user study.**

**Rating:** 7
**Confidence:** 5

**Review:**

### Summary
The manuscript documents a significant group effort to push toward better metadata documentation for machine-learning datasets, tackling important problems faced by the community. The resulting metadata format, Croissant, has already seen considerable adoption in practice.

While the basic architecture of Croissant is well thought-out, this does not hold to the same degree for the Responsible AI extension also presented in the paper. However, I understand that this extension might still be under active development, and I would encourage the authors to rethink their design choices here to enable greater standardization and machine readability.

This submission could be a clear accept, but it currently faces two problems.
First, the user study is flawed (see "Correctness" for more details and recommendations).
Second, although motivated by ML-specific challenges, the paper lacks clarity and detail on how Croissant tackles which specific challenges that are inadequately addressed by existing tools.
If these problems are fixed, I will support the publication of this manuscript on the NeurIPS Datasets and Benchmarks track.

### Strengths
- (S1) Importance of problem(s) tackled (lack of structured metadata for ML datasets and lack of metadata schemas meeting the needs of ML research and practice)
- (S2) Basic architecture (building on schema.org), interoperability with existing standards, ecosystem of tools and integrations
- (S3) Considerable pre-publication adoption in practice
- (S4) Attempt to evaluate the proposed metadata format by conducting a user study

### Weaknesses
- (W1) Realization of the user study (all of it: setup, implementation, evaluation, presentation)
- (W2) Design of Responsible AI extension (appears more ad-hoc and less well thought-out than the rest of the metadata format; effectively not machine-readable; could benefit from structured vocabulary)
- (W3) Motivation/presentation mismatch (you motivate your work with the need for ML-specific metadata formats but a large part of your exposition, including the case study, focuses on metadata that is pretty standard in other domains as well)

**Strengths:**

See above.

**Additional Feedback:**

- Out of curiosity: Why the name "Croissant"?
- The rate of 25% invalid Croissant files you report in the appendix seems high. Can you comment on that?
- Grammar: Please use the correct possessive forms when referring to non-human entities (e.g., "Data's prominent role" -> "The prominent role of data"; "datasets' attributes" -> "the attributes of datasets"); if that breaks your layout, you can condense your writing in other places to stay in the page limit
- Typos, missing words, etc.: I recommend an additional editorial pass over the manuscript
- Abbreviations: Make sure you define all your abbreviations before their first usage (e.g., "RAI" is used before it is defined)
- Table and figure layouts: You might want to use subfigures with subcaptions here.
- When you write "golden data", you probably mean "ground-truth data".
- Can you update reference [21]?

**Clarity:**

The paper could explain more clearly the value added by Croissant to address ML-specific data-documentation challenges.

**Correctness:**

On the user study (Section 4): I appreciate that you set out to conduct a user study, but there are so many things wrong about it that I do not even know where to start. I would recommend removing it from the paper completely (that way, you would have more space to emphasize and explain the ML-specific features of Croissant) or redoing it from scratch following state-of-the-art user-study procedures.

Here is a sample of questions and concerns your user study raised for me:

### Setup and Implementation
- The Croissant attributes to annotate have nothing to do with the ML-specific challenges you set out to address, which limits the insights we can gain from the user study.
- How did you recruit the annotators? If they had prior contact with the working group and knew that they were asked to evaluate a tool designed by those running the study, the responses will be biased.
- What are the annotator demographics?
- Why do you let annotators record their own timings? This problem has a technical solution.

### Evaluation
- How many annotations are there? Sometimes you say 29, sometimes you say 30.
- Why do you count each dataset annotation as a separate observation, allowing the same person to have a different impact on the evaluation, depending on how many datasets they annotated? This way, your observations are not independent, and you are blowing up your numbers. Your description also wavers between "participants" and "annotations", treating the two as equivalent when they are not.
- Evaluation with BLEU for categorical attributes does not make sense to me. Can you explain why you did that?
- The fact that you feel like you have to evaluate with BLEU could point to some shortcomings in the design of your metadata format (especially for the RAI extension, as mentioned already under W2).
- Maybe I missed it, but what is the value of n for your n-grams?
- Your BLEU scores seem pretty low, even for things that seem straightforward to annotate (like "datePub"). Can you comment on that (maybe this just shows that BLEU is not the right metric here)?

### Presentation
- You have such small n here – show observation counts, not percentages.
- Align figures with identical semantics on their x axis and their y axis (Figures 4–7) to enable visual comparisons without misleading readers.
- The colors you chose for Figures 4–7 have no intuitive mapping to the underlying ordinal data, and therefore, they hurt more than they help. Please use a discretized perceptually-uniform sequential colormap (if you want to use color) or simply do away with the color altogether (it does not really add much here).

(These points also hold for the figures in your appendix. Plus: (1) Why the binning and pie-charting in Figure 14? You have so few observations that you can show them, e.g., in a strip plot [splitting by dataset and showing time on a continuous axis]. (2) I am pretty sure that you can find a more effective way to visualize the information in Figures 9 and 11 than using radar charts.)

**Documentation:**

If you decide to include the user study in the revised version:
Please make the disaggregated annotations and their associated evaluation data available as a research-data artifact (depositing it with a unique DOI), and also document the missing metadata I asked about in the Correctness section.

**Ethics:**

The fact that the Croissant format was recommended by the track to which this paper was submitted suggests that there was some personnel overlap between the Croissant working group and the decision-makers on the track, but I do not think that this raises ethical concerns, especially given the adoption of the format also by other platforms.

**Limitations:**

The authors discuss the limitations of their user study in a separate subsection of Section 4, and they mention some additional limitations and avenues for future work in Section 5.
I see some more limitations for the user study (see "Correctness"), and the paper would also benefit from discussing which ML-specific needs Croissant doeos not (yet) address (in full or in part).

**Opportunities For Improvement:**

See above and "Correctness".

**Relation To Prior Work:**

The discussion of related work is rather shallow.
For example, at the end of page 2, you write (referring to related efforts) that "these standards do not entirely meet the specialized needs of data management within the ML domain."
I agree with that, but the write-up would benefit from being more explicit about what is missing and how Croissant helps fill that gap.
You might also want to mention FAIR data principles in the introduction already since they provide context for your work.

**Summary And Contributions:**

The paper introduces Croissant, a metadata format for dataset documentation designed to improve the discoverability, portability, reproducibility, and interoperability of machine-learning datasets that has already been widely adopted in practice.

---

> ### Author Rebuttal · Authors · 2024-08-21
>
> We thank the reviewer for their thorough review and are encouraged that they recognize the importance of the problems our work addresses.
>
> Below, we address the specific points and questions raised in the review:
>
> __User Study: Setup and Implementation__
> 1. __“The Croissant attributes ... from the user study.”__
>
> We believe that the challenges of ML practitioners are addressed by Croissant as a whole, i.e., by all the layers of the format, and not just by ML-specific properties. The user study evaluates a representative subset of Croissant properties, giving us an understanding of how users perceive Croissant when annotating ML datasets. We believe the study does inform the format's usability from the dataset author perspective, and allows us to highlight challenges we can address with future releases of the Croissant.
>
> 2. __“How did you recruit ... the responses will be biased.”__
>
> We recruited participants from Croissant’s large mailing list. Most participants (all except three) are not directly involved in the development of Croissant. To minimize annotator variance, we collected three annotations per dataset. In future evaluations, we plan to expand the study to include a broader group of annotators, datasets, and practitioners not previously involved in Croissant.
>
> 3. __“What are the annotator demographics?”__
>
> We collected demographic information from all annotators and made it openly accessible in the User research study available here [1].
>
> 4. __“Why do you let annotators record their own timings? This ...”__
>
> We were not able to implement that in this first study. However, we appreciate this suggestion and will consider it in a follow-up evaluation.
>
> __User Study: Evaluation__
> 1. __“How many annotations are there?”__
>
> Thank you for pointing out this discrepancy. We received 30 annotations and will correct the numbers throughout the paper.
>
> (Note: The user research report at [1] shows responses for 32 annotations since we received 2 delayed annotations. We added those to the report but not to the paper to avoid any confusion).
>
> 2. __“Why do you count each dataset annotation as a separate observation...”__
>
> We agree that using “participants” and “annotations” interchangeably is confusing and will streamline the terminology in the final paper. We updated the Figures (see attached pdf) as well as the text to reflect “Percentage of Annotations” instead of “Percentage of Annotators” as you correctly pointed out.
>
> Moreover, we have made the User research study publicly available here [1].
>
> 3. __“Evaluation with BLEU for categorical attributes does not make sense to me. Can you explain why you did that?”__
>
> We used BLEU exclusively for attributes requiring free-form text input, not for categorical attributes.
>
> 4. __“The fact that you feel like ...”__
>
> Following earlier machine-readable vocabulary schemas, Croissant includes a mix of free-text attributes and attributes with predefined values. BLEU was chosen to address the current reliance on free-text attributes in Web vocabularies such as schema.org, which also applies to the initial release of Croissant.
>
> 5. __“Maybe I missed it, but what is the value of n for your n-grams?”__
>
> We used the default BLEU calculation value of n equal to 4.
>
> 6. __“Your BLEU scores seem pretty low.."__
>
> We briefly mention this phenomenon but will clarify it further in the final version (see “Attributes’ average BLEU scores also diverge based on their expected values...” in Section 4). Even for attributes like datePub, the variety of date formats impacts the BLEU score. In contrast, attributes with predefined values, such as Language or URL, have an average score of 0.59.
>
> __Relation to Prior Work__
> 1. __“The discussion of related work... context for your work.”__
>
> Due to the space limit for the submission, we had to condense the related work section. In the camera-ready version, we will extend the related work section building on related dataset toolkits which we discuss in our publication on the Croissant RAI extension [2].
>
> __Documentation__
> 1. __“If you decide...Correctness section.”__
>
> Thank you for pointing this out. As suggested, we have compiled a user study report and made it publicly available here [1] with a unique DOI.
>
> __Responsible AI Extension__
> 1. __“While the basic...machine readability.”__
>
> While we briefly introduced the RAI extension to provide a complete picture of the current state of the Croissant format, it has been described in detail in a separate publication [2].
>
> __Further Feedback__
> 1. __“Out of curiosity: Why the name ‘Croissant’?”__
>
> Similar to a croissant, which consists of multiple layers of dough, the Croissant format is organized around four layers: the Dataset Metadata layer, the Resources Layer, the Structure Layer, and the Semantic Layer. We also came up with this name during an early workshop in Paris, where croissants are a favorite breakfast fare.
>
> 2. __“The rate of 25%...”__
>
> The rate of invalid Croissant files is due to authentication issues while trying to access private datasets. We use [Croissant Health](https://github.com/mlcommons/croissant/tree/main/health) to monitor the health of the Croissant ecosystem by crawling online JSON-LD files shared across repositories. Currently, Croissant Health performs this check for Hugging Face and OpenML datasets, but will be extended in future to further repositories. See https://github.com/mlcommons/croissant/blob/main/health/visualizer/report_openml.ipynb for further details.
>
> We also thank the reviewer for highlighting issues related to grammar, typos, abbreviations, and clarity (e.g., gold vs. ground truth data), which we have addressed in the updated version of the paper.
>
> References:
>
> [1] Croissant Working Group. (2024). Croissant - User Research Report (0.1). Zenodo. https://doi.org/10.5281/zenodo.13350974
>
> [2] Jain, Nitisha, et al. "A Standardized Machine-readable Dataset Documentation Format for Responsible AI." arXiv preprint arXiv:2407.16883 (2024).

---

> > ### Comment · Reviewer_fBtg · 2024-08-22
> > **Response to rebuttal comments**
> >
> > Thank you for responding to the questions and concerns raised in the review process.
> > I especially appreciate the graphical abstract you provided as part of the rebuttal PDF.
> >
> > Regarding the updated figures, I would recommend that you harmonize the styling and fonts to be consistent across all figures and match the fonts of the paper.
> > Regarding the user study, I encourage you to underline the preliminariness of the design and evaluation.
> > Although one could argue that some user study is better than none, I am concerned that the ML community, where user studies are not yet common, will take your user study as precedent to design future user studies – even though your study is far from reflecting the state of the art.
> > To mitigate this risk, it should be very clear from the write-up that the paper got through despite (and not because of) the user study, and that future work must be more rigorous when designing and implementing user studies.
> >
> > Since I recognize your responsiveness to reviewers' feedback, I will raise my score to "Accept", even though my concerns with the user study remain.

---

> > > ### Author Response · Authors · 2024-08-23
> > >
> > > We would like to thank the reviewer for their response, and are grateful for the score update. We will harmonize fonts and styles in the camera ready version, and will make the shortcomings of the user study as clear as possible.

---

### Official Review · Reviewer_e6ro · 2024-07-25
**A combined description of the croissant metadata standard and a user study of annotating datasets**

**Rating:** 7
**Confidence:** 4

**Review:**

I found this paper surprisingly difficult to read. I think one of the big challenges is the lack of space that's available to the authors. The power of a machine readable and human readable metadata standard is very high. The challenge is that without a couple of example datasets to follow along it's a very dry topic and the hard work that has gone into developing the standard does not come across.

I thought the user study was good and I felt it was helpful to understand specifically where users were agreeing or disagreeing around the evaluation criteria (completeness, conciseness, readability and understandability). For example the Commonly confused attribute pair of sc:creator and sc:publisher coming from an existing schema.org vocabulary is helpful to know and can give users a croissant extra context to avoid this pitfall in the future.

I don't think that the BLEU scores added much to the paper. I understand wanting to evaulate the actual documentation and how similar they are but I think that needed to be done by expert humans rather than a text comparison algorithm. The limitations of the method are clearly explained: the BLEU score is driven mostly by the style and length of answer expected rather than the content itself. Two date times will be much more similar to each other than long form text descriptions.

Ultimately I think the implementation of croissant is clearly very powerful but this paper did not describe the benefits as well as it could have done.

**Strengths:**

I appreciated the figures two and three that show what the machine readable metadata files look like. I also appreciated the section on related work and the implementation of a usability study.

**Additional Feedback:**

Thank you for all the hard work on croissant!! I know how much effort it takes to align machine readable metadata with human understandable descriptions.

**Clarity:**

The paper is as well written as it can be given my earlier point that I think this paper is trying to squeeze too much into a short page limit. Explaining the metadata standard would be sufficient on its own, but maybe it makes sense to focus this on the user testing study and point the reader to documentation and examples for croissant elsewhere?

**Correctness:**

I think the BLEU scores are misleading and could lend themselves to misinterpretation. But the method is clearly described and the broader claims in the paper are well justified.

**Documentation:**

The supplementary material has clear documentation for how to use croissant and also how the user study was undertaken.

**Ethics:**

No ethical concerns

**Limitations:**

The limitations for the user study are described. The limitations to croissant are not.

**Opportunities For Improvement:**

My recommendation to improve this paper would be to enhance the examples with more context about the data they are describing. I would also recommend reevaluating the comparison of the attributes provided in the user testing study.

Maybe my biggest piece of feedback is that this paper can either introduce croissant or describe a user testing study. The page limit makes it difficult to do both in one submission.

**Relation To Prior Work:**

Yes - this was well explained.

**Summary And Contributions:**

This paper introduces croissant - a metadata format for datasets that creates a shared representation across ML tools, frameworks and platforms. The authors give an overview of related work makes a clear argument for the need for this metadata format.

The paper describes the croissant format At the data set metadata layer, the resources layer, the structure layer, and the semantic layer. They also explore the responsible AI extension to croissant and describe some croissant tools and integrations.

The authors have undertaken a evaluation of croissant through a user study with machine learning practitioners. They describe the recruitment of nine volunteers from the croissant development community who completed metadata definitions from an average of three data sets each and answered questions about their experience of undertaking the task. The authors manually studied the provided answers and assess the agreements of participants. To measure agreement they calculated BLEU scores between attribute annotations and categorised the user experience questions across 5 dimensions: consistency, completeness, conciseness, readability, and understandability

---

> ### Author Rebuttal · Authors · 2024-08-21
>
> We are thankful to the reviewer for their constructive feedback and comments. Our responses are inline with the comments below -
>
> __Review__
> 1. __“The challenge is that without a couple of example datasets to follow along it's a very dry topic and the hard work that has gone into developing the standard does not come across.”__
>
> We described working with the PASS dataset in Sections 3.2 to 3.4 of the manuscript but couldn’t describe more datasets due to space constraints. Additionally, we have provided code examples in the Appendix 6.1.1. In the camera ready version, we will add more examples in the Appendix. Also, we recommend readers to check the Croissant Github repository for more examples [1].
>
> [1] ​​https://github.com/mlcommons/croissant/tree/main/datasets
>
> 2. __“I don't think that the BLEU scores added much to the paper. I understand wanting to evaulate the actual documentation and how similar they are but I think that needed to be done by expert humans rather than a text comparison algorithm.”__
>
> We used BLEU exclusively for attributes requiring free-form text input. In the future, we aim to extend this evaluation over a larger sample size and a manual comparison based on a gold standard.
>
> 3. __“Ultimately I think the implementation of croissant is clearly very powerful but this paper did not describe the benefits as well as it could have done.”__
>
> We are thankful to the reviewer for their comment and welcome any suggestions how we can improve the format’s presentation in addition to the comments provided above as well as suggested by other reviewers. Moreover, we will include a visual abstract in section 1 (see attached pdf) to make it easier to obtain an overview of the scope, features and integrations of the project.
>
> __Opportunities For Improvement__:
>
> 1. __“Maybe my biggest piece of feedback is that this paper can either introduce croissant or describe a user testing study. The page limit makes it difficult to do both in one submission.”__
>
> We agree with the reviewer that the restricted page size makes it difficult to present both the format and the user study in depth. However, in the final paper version we will have one more page which allows us to discuss some aspects of the format in more detail.
>
> On the other hand, introducing just a design artifact (i.e., Croissant) without any evaluation would have raised some questions. We think it's important to not just present the solution to the problems we discuss in the introduction of the paper and rely on adoption and impact figures as a measure of how good the format is, but also to discuss how users understand and use the proposed format, at least to some extent. In future work, we plan to build on this work to conduct a large-scale user study. Also, the current user research study has been made available here [1].
>
> [1] Croissant Working Group. (2024). Croissant - User Research Report (0.1). Zenodo. https://doi.org/10.5281/zenodo.13350974
>
> __Limitations__:
> 1. __“The limitations for the user study are described. The limitations to croissant are not.”__
>
> We will include a description of Croissant’s current limitations in the camera-ready version.

---

> > ### Comment · Reviewer_e6ro · 2024-08-31
> > **Thank you - and no change in score**
> >
> > With my apologies for such a late reply (at the end of the summer vacation!) I acknowledge the response from the authors.
> >
> > Although I think the clarifications are fair, and I like the visual abstract very much, I think my concerns around the BLEU scores remain high. I think the work is valuable and I am happy with the scores I originally provided.
> >
> > I look forward to citing the larger usability study :)

---

### Official Review · Reviewer_gKdc · 2024-07-25
**Promising Dataset Metadata Standard with Impressive Format, Limited Study**

**Rating:** 8
**Confidence:** 4

**Review:**

The paper presents a well-thought-out and potentially impactful approach to standardizing ML dataset metadata. The layered structure and extensibility of Croissant show significant promise for wide adoption in the ML community. However, there are concerns about the format's complexity and some limitations in the evaluation study.

## Pros:
- Addresses a critical need for standardized dataset metadata in ML
- Builds on existing standards (schema.org) for broader compatibility and interoperability
- Significantly improves dataset discoverability, portability, and interoperability
- Already adopted by major platforms (Hugging Face, Kaggle, OpenML), indicating real-world applicability
- Includes a Responsible AI extension, addressing emerging ethical concerns in ML
- Provides robust implementation tools (Python library, Croissant Editor)
- Supports file sequences and regex matching, enhancing flexibility for complex datasets
- Offers a comprehensive approach to describing various data types used in ML

## Cons:

### Regarding Croissant
- Complex structure may challenge human readability/writability, potentially hindering adoption
- Regarding the data format, implicit mappings could lead to unexpected behaviors if users are not fully aware of class properties
- Nested structures, while expressive, may be difficult to validate and process efficiently in big datasets
- An example of handling complex data structures (e.g., HDF5, Zarr, DICOM) is missed.

### Regarding the study
- Limited evaluation scope (10 datasets, 9 participants) and potential participant bias due to the participants origins (Croissant community)
- Lacks comparative analysis with other existing metadata standards
- The study fails to explore non-expert usage or real-world scenarios sufficiently
- Use of BLEU scores for agreement measurement is innovative but may not be ideal for all attribute types as reported with in the RAI-related results (Figure 5: conciseness). Considering the sample size, a manual comparison with a gold standard would be more robust
- Results from the study are not shared openly or could not be found

**Strengths:**

- Addresses a significant and growing need in the ML community for standardized dataset descriptions
- Highly extensible and interoperable design, building on established web standards
- Early adoption by major ML platforms indicates strong potential for widespread impact
- Thoughtful consideration of responsible AI principles through dedicated extensions
- Comprehensive approach to describing various data types and structures used in ML
- Provision of tools like the Croissant Editor to facilitate adoption and reduce friction

**Additional Feedback:**

- Consider implementing standardized value sets for fields like licenses (e.g., SPDX: https://spdx.org/licenses/ ) to ensure consistency
- Develop semantic search capabilities specifically for Croissant metadata files to enhance discoverability
- Explore and document versioning strategies for different metadata layers to support dataset evolution

**Clarity:**

The paper is generally well-written and structured. However, it could benefit from more detailed examples of complex use cases and real-world applications to illustrate the full capabilities of the Croissant format.

**Correctness:**

The claims made about the Croissant format appear correct and well-founded. However, the limited scope of the evaluation study means that broader validation is needed to fully support all claims about usability and effectiveness across diverse ML scenarios.

**Documentation:**

The paper provides sufficient detail on the format specification and available tools. In parallel, the project has a good documentation hoster on its repository and an active community supporting the adoption of the format. However, more guidance on extending the format for specific domains or use cases would be beneficial.

**Ethics:**

No major ethical concerns are apparent. The inclusion of a Responsible AI extension shows thoughtful consideration of ethical issues in ML data management. However, more detailed guidelines on ensuring ethical data use through metadata could further strengthen this aspect.

**Limitations:**

The authors acknowledge some limitations of their study, but could further address:
- Potential challenges and strategies for widespread adoption, especially among non-experts
- Detailed exploration of handling very large or complex datasets
- Performance implications of using Croissant in ML pipelines, including any potential overhead
- Limitations of the current evaluation study, including participant bias and limited dataset variety

**Opportunities For Improvement:**

- Significantly expand the evaluation with a larger, more diverse participant group, including non-experts
- Include a thorough comparative analysis with other metadata standards and approaches
- Provide detailed case studies on large-scale, complex datasets (e.g., ERA5: https://github.com/google-research/arco-era5 ) to demonstrate scalability
- Develop community guidelines and repositories for extending the semantic layer with domain-specific needs. Consider using the RAI extension as an example on how to do this
- Improve human-readability, possibly by offering a YAML conversion option as an intermediate format
- Address data provenance more comprehensively and demonstrate handling of complex data structures
- Conduct a real-world evaluation using unstructured datasets (e.g., from Zenodo) to test format flexibility
- Explore versioning capabilities for different metadata layers to support evolving datasets

**Relation To Prior Work:**

While the paper builds on existing standards like schema.org and is inspired by Model cards, it lacks a comprehensive comparative analysis with other metadata approaches in the ML field (especially previous attempts to use semantic technologies in open datasets, e.g., https://opendata.swiss/en). This comparison would help contextualize Croissant's innovations and advantages.

**Summary And Contributions:**

This paper introduces Croissant, an innovative metadata format for ML datasets built on JSON-LD and schema.org. It aims to significantly improve dataset discoverability, portability, and interoperability across ML platforms.  Key contributions include:

1. A layered metadata structure (Dataset Metadata, Resource, Structure, and Semantic layers)
2. Compatible with extension for field-specific applications. i.e. Responsible AI extensions addressing ethical concerns
3. Integration with major ML repositories (Hugging Face, Kaggle, OpenML)
4. Tools for implementation (Python library, Croissant Editor)
5. Open-source development with an active GitHub Community.
6. Enhance the Open Research Data and FAIR (Findable, Accessible, Interoperable, and Reproducible) ecosystem.

The format shows great promise in standardizing ML dataset descriptions, simplifying ML workflows, and enhancing reproducibility and data sharing.

As a remarkable note, It is impressive that the format has been widely adopted by the leading open ML dataset providers, which reflects a tight connection with current ML practices and an active and diverse community behind the project.

---

> ### Author Rebuttal · Authors · 2024-08-21
>
> We are thankful to the reviewer for their feedback and comments. Our responses are inline below.
>
> __Regarding Croissant__
> 1. __“Complex structure may challenge”__
>
> We acknowledge reviewer's comment. There is some inherent complexity because of the diversity of (ML) datasets, and the level of description needed to make datasets directly usable for machine learning. We strived to create an understandable conceptual model by organizing the constructs of the format into independent layers.
>
> We believe that effective tools and comprehensive documentation are the primary way to make Croissant datasets easy for users to grasp and use, but we acknowledge that our tools are currently work in progress.
>
> 2. __“... implicit mappings could lead to unexpected behaviors”__
>
> Mapping dataset contents to external vocabularies is an advanced functionality of Croissant. We tried to make it as easy and natural as possible for users by introducing implicit mappings. Indeed, these can lead to unexpected behavior, as pointed out by the reviewer. We plan to add functionality to the Croissant validation library to clearly highlight to users whenever an implicit mapping is happening, with a clear link to the mapped classes/properties documentation, so that they are aware of the consequences. We are open to additional suggestions on how to make mappings easy to use and understandable.
>
> 3. __“Nested structures, while ”__
>
> We agree with the reviewer's comment. However, nested structures are required to represent the contents of a large number of important  ML datasets. While there is still a lot of room to develop efficient data loading libraries for Croissant, in the short run, we recommend the following best practices for efficient loading of big datasets-
> - Using RecordSets that are as "flat" as possible
> - Using random-access-compatible data formats for the underlying FileSet/FileObject, such as Parquet
> - Using a simple structure of FileSet/FileObject
>
> Some libraries, like TFDS and Hugging Face, preprocess datasets to convert them to formats that enable efficient loading. In the long run, we would like to investigate doing that automatically based on Croissant descriptions of unoptimized datasets (e.g., because of nesting). Moreover, we are working on improving our tools and documentation to make the above recommendation evident for new users.
>
> 4. __“handling complex data structures ..."__
>
> We are actively working on a GeoSpatial extension for Croissant (named Geo-Croissant) and will be generating examples of handling HDF5, Zarr file formats, as geospatial data is mostly stored in these file formats.
>
> __Regarding the study__
> 1. __“Limited evaluation scope”__
>
> We agree and plan to undertake larger user studies in future. With this study, we aimed to present an initial usability evaluation that provides a better understanding of how users can annotate common machine learning datasets using the Croissant format and its documentation for a subset of Croissant attributes.
>
> 3. __“Lacks comparative analysis”__
>
> Could the reviewer please clarify which standards they are referring to? If it is dataset documentation toolkits such as Dataset Cards, Crowdworksheets, and Kaggle metadata, we can refer to our publication on the Croissant RAI extension [1].
>
> To develop the RAI extension, existing dataset documentation vocabularies were thoroughly examined to identify overlaps and gaps in relation to the Croissant vocabulary, i.e. the attributes. An overview of these vocabularies is presented in Table 1 of our RAI publication [1].
>
> 4. __“The study fails to explore..”__
>
> Most of the Croissant use-cases are not for technical users, for eg. as defined in the Croissant RAI specification [2] and “Use Cases” section of [1]. These use-cases target some non-technical users, for eg. regulators or AI compliance teams in organizations. Additionally, several attributes in Core Croissant vocabulary such as “name”, “description”, “cite as” are relevant to the non-technical audience and are actually overlapping with data cards, so they'd be created automatically from data cards. Apart from this, most new attributes in Croissant are for loading datasets in multiple ML frameworks (technical audience) or discoverability (technical audience).
>
> 5. __“Use of BLEU scores..”__
>
> We agree and in future, we aim to extend this evaluation over a larger sample size and a comparison based on a gold standard.
>
> 6. __“Results from the study..”__
>
>  The User study has been made openly accessible here [3].
>
> __Opportunities For Improvement and Additional Feedback__
>
> We are already working on some of the suggestions for the future Croissant releases.
>
> 1. __“Consider implementing standardized..”__
>
> Some of the licenses such as SPDX are defined in the [Croissant specification](https://docs.mlcommons.org/croissant/docs/croissant-spec.html#required). Additionally, we are looking at these licenses and working with teams like the Data Provenance Initiative to understand which licenses from the Linux Foundation list apply to ML datasets. Also, going forward, we plan to pre-define ranges for a number of other attributes to enhance data consistency.
>
> 2. __“document versioning strategies..”__
>
> The Croissant specification covers versioning in more detail than we could include in the paper, with support for "live" and "snapshot" datasets. We agree with the reviewer that there is room for much more detailed and granular versioning strategies, and plan to investigate that in the future if this turns out to be an important functionality for Croissant users.
>
> References:
>
> [1] Mubashara Akhtar et al. Croissant RAI Specification. Technical report, 2024. https://docs.mlcommons.org/croissant/docs/croissant-rai-spec.html
>
> [2] Jain, Nitisha, et al. "A Standardized Machine-readable Dataset Documentation Format for Responsible AI." arXiv:2407.16883 (2024).
>
> [3] Croissant Working Group. (2024). Croissant - User Research Report (0.1). Zenodo. https://doi.org/10.5281/zenodo.13350974

---

### Official Review · Reviewer_96aN · 2024-07-25
**clear contributions and enjoy reading this work**

**Rating:** 8
**Confidence:** 5
**Correctness:** Yes
**Clarity:** Yes

**Review:**

The paper’s pros are:

	1.	Innovative Metadata Standard: The introduction of a standardized metadata format that addresses significant challenges in ML data management, including discoverability, portability, and interoperability, is well-articulated (Section 1).
	2.	Broad Adoption and Integration: The format’s integration with major dataset repositories like Hugging Face, Kaggle, and OpenML, as well as its support by Google Dataset Search, demonstrates its practical applicability and widespread acceptance (Section 3.6).
	3.	Comprehensive Evaluation: The user study involving ML practitioners provides valuable insights into the usability and effectiveness of the Croissant format, highlighting its strengths in readability and understandability (Section 4).

The paper’s cons are:

	1.	Limited Discussion on some responsible evaluations like Fairness Testing: The paper does not explicitly address fairness testing or potential biases in the metadata format, which is critical for responsible AI practices (Section 4.3).
	2.	Complexity for New Users: While the Croissant Editor tool is mentioned, creating and modifying dataset descriptions may still be challenging for users unfamiliar with the format, potentially limiting its adoption (Section 3.6).

**Strengths:**

The paper makes a significant contribution by introducing a standardized metadata format that addresses key challenges in ML data management. The Croissant format enhances the discoverability, portability, and interoperability of ML datasets, supporting responsible AI practices and fostering collaboration within the ML community. T

**Additional Feedback:**

N.A.

**Documentation:**

Yes

**Ethics:**

Yes

**Limitations:**

How do we connect this form to Industri's catalog from databricks and snowflake?

if we can do that, many industry engineers will definitely benefit a lot

**Opportunities For Improvement:**

Maybe we can add some section to tell users if there are any safety/fariness/privacy/etc concerns of the paper?


Enhance the Croissant Editor tool to provide more intuitive, guided workflows for users unfamiliar with the format. Additionally, offer more detailed tutorials and documentation to help users get started with creating and modifying metadata descriptions. I have not seen any figures related to that editor tool

**Relation To Prior Work:**

Yes

**Summary And Contributions:**

Croissant aims to create a shared representation for ML datasets, enhancing their usability and facilitating responsible AI practices. The format is supported by several popular dataset repositories, enabling easy loading into common ML frameworks. The paper also presents an initial evaluation showing that Croissant metadata is considered readable, understandable, complete, and concise by human raters.

---

> ### Author Rebuttal · Authors · 2024-08-21
>
> We thank the reviewer for their constructive feedback and comments. Our responses to specific questions are inline below:
>
> 1. __“... responsible evaluations like Fairness Testing”__
>
> While we briefly introduced in the paper the RAI extension, it has been described in detail in a separate publication [1]. This paper focuses on the Core Croissant vocabulary.
> In the paper, we mentioned that Croissant addresses FAIRness testing as a use-case using the Responsible AI (RAI) extension (Refer to Section 3.5 - “The Croissant-RAI Extension” and [1] for more details).
>
> [1] Jain, Nitisha, Mubashara Akhtar, Joan Giner-Miguelez, Rajat Shinde, Joaquin Vanschoren, Steffen Vogler, Sujata Goswami et al. "A Standardized Machine-readable Dataset Documentation Format for Responsible AI." arXiv preprint arXiv:2407.16883 (2024).
>
> __“... potential biases in the metadata format”__
>
> These are the measures we took to address potential biases while designing the Croissant format:
>
>  - We designed the Croissant format  in an open, participatory way.
>     - The Croissant Working Group (WG) consists of diverse stakeholders and domain experts coming from academia, industry, research organizations and other collaborative networks such as AI for Public Good network, and is open for anyone to participate.
>     - Use cases were discussed and presented to the WG members  (including domain experts) as they were developed to ensure covering diverse views and priorities.
>  - The schema is designed to be modular and extensible to accommodate domain specific attributes and concerns in the design of the Core Croissant format.
>  - Finally, we continuously collect feedback from working group members, and users (such as NeurIPS D&B Track authors), and are committed to incorporating that feedback in future versions of Croissant.
>  - Additionally, Croissant is based on schema.org, which is an established vocabulary. We Surveyed existing vocabularies for dataset description (including RAI ones for the extensions) and strived to use commonly adopted terminology and optimize for understandability to minimize biases.
>
> 2. __“Complexity for New Users” and Opportunities For Improvement__:
>
> We agree that Croissant will take time to mature. To improve documentation for users, we plan to create a Croissant website as a  “landing page” with examples, reproducible notebooks, and step-by-step instructions to use the  Croissant Editor. We also plan to improve the user interface of the Croissant Editor.
>
> 3. __“...Connection with databricks and snowflake”__
>
> At this point we do not have examples showing connection of Croissant with cloud-based data warehousing platforms as Databricks or Snowflake, but we are open to collaborating with these organizations if they see value in supporting Croissant.

---

> > ### Comment · Reviewer_96aN · 2024-08-28
> >
> > sounds good to me

---

### Author Rebuttal · Authors · 2024-08-21

We sincerely thank all reviewers for their thorough and constructive feedback. We are encouraged by the positive reception of Croissant.

Reviewers recognized the benefit of standardizing metadata for ML datasets and appreciated our approach, noting the well thought-out layered structure and extensibility of Croissant. The broad adoption and integration with major dataset repositories and ML frameworks are seen as significant strengths. Reviewers also commend Croissant's foundation on schema.org, which enhances interoperability. We are grateful for this positive reception, which aligns with our goals for Croissant and reflects our commitment to creating a practical, widely applicable solution for ML dataset metadata.

The reviewers also raised several shared questions.

Firstly, reviewers pointed to potential challenges for new users to adopt  the format. We recognize this concern and are continuously working to improve documentation and tooling, particularly the Croissant Editor. We will also include in the appendix additional excerpts of examples from our code repository that show how users can get started with Croissant.

Secondly, reviewers expressed concerns about the scope and presentation of the user study. We acknowledge these limitations. We will expand the discussion of the limits of this initial usability evaluation in Section 4.3 and make sure the distinction between annotators and annotations is crystal-clear in the text and reflected in the figures (see attached pdf).

Thirdly, reviewers suggested improvements in the presentation. We agree and will copy-edit the manuscript, including a visual abstract in section 1 (see attached pdf) to make it easier to obtain an overview of the scope, features and integrations of the project.

We are committed to integrating the reviewer suggestions in the paper and in our ongoing development of Croissant. We believe these improvements will further strengthen Croissant's contribution to the ML community by enhancing dataset discoverability, portability, and responsible use. We again thank the reviewers for helping us make the format and presentation of Croissant clearer. Additional, detailed responses can be found in our direct comments to reviewers.

---

### Decision · Program_Chairs · 2024-09-26

**Decision:**

Accept (Spotlight)

**Comment:**

This paper introduces Croissant, a metadata format for ML datasets. It aims to create a shared representation for ML datasets, enhancing their FAIR-ness. The format is supported by several popular dataset repositories, enabling easy loading into common ML frameworks. The paper also presents an initial evaluation showing that Croissant metadata is considered readable, understandable, complete, and concise by human raters.
The reviewers are in agreement to accept the paper, while provide with useful comments to further improve the paper. Authors and reviewers engaged in a discussion during the rebuttal phase, and  authors are committed to implement several recommendations in the final paper version.
I concur with the reviewers that this is a very important contribution towards metadata standards for ML datasets.